# Eating Fermented: Health Benefits of LAB-Fermented Foods

**DOI:** 10.3390/foods10112639

**Published:** 2021-10-31

**Authors:** Vincenzo Castellone, Elena Bancalari, Josep Rubert, Monica Gatti, Erasmo Neviani, Benedetta Bottari

**Affiliations:** 1Department of Food and Drug, University of Parma, Viale delle Scienze, 49/A, 43124 Parma, Italy; elena.bancalari@unipr.it (E.B.); monica.gatti@unipr.it (M.G.); erasmo.neviani@unipr.it (E.N.); benedetta.bottari@unipr.it (B.B.); 2Food Quality and Design Group, Wageningen University, Bornse Weilanden 9, 6708 WG Wageningen, The Netherlands; josep.rubert@wur.nl

**Keywords:** lactic acid bacteria, fermented foods, health benefits, bioactive compounds

## Abstract

Lactic acid bacteria (LAB) are involved in producing a considerable number of fermented products consumed worldwide. Many of those LAB fermented foods are recognized as beneficial for human health due to probiotic LAB or their metabolites produced during food fermentation or after food digestion. In this review, we aim to gather and discuss available information on the health-related effects of LAB-fermented foods. In particular, we focused on the most widely consumed LAB-fermented foods such as yoghurt, kefir, cheese, and plant-based products such as sauerkrauts and kimchi.

## 1. Introduction

Lactic fermented foods have been consumed by humankind since ancient times for their organoleptic characteristics and prolonged shelf-life. Nowadays, the consumption of fermented foods is also driven by a health-related perspective. The market of fermented foods is nowadays touching all countries and shows an increasing trend [1]. Natural microflora of fermented foods is often composed mainly by LAB. LAB are Gram-positive, nonsporing, generally nonmotile, with complex nutritional requirements, depending on the presence of a fermentable carbohydrate for active growth. As an end product of this fermentation, LAB produce copious amounts of lactic acid alone (homofermentative) or together with acetic acid, ethanol, and carbon dioxide (heterofermentative) [2]. Genetic diversity and habitat variation are considerably wide within LAB. Therefore, no general limits for pH, a_w_, temperature, or other parameters exist for the LAB, and the growth-limiting circumstances vary depending on the considered species (Table 1) [3]. LAB are responsible for a great diversification in the flavor and texture of fermented foods, and can be, in some circumstances, responsible for food spoilage [4]. They can also release an array of health-modulating compounds and signal molecules in the matrix during fermentation. These food-derived bacteria and their metabolites can interact with the intestinal microbiome and with the host itself like members of an orchestra playing a health symphony for the intestine and the organisms in general. Regular ingestion of fermented food can therefore contribute in many ways to homeostasis and organism functions. A link between health and the use of eating LAB-fermented foods has been hypothesized since Metchnikov’s intuition that an increased life expectancy of balcanic population was attributable to the significant consumption of lactic fermented milk. Since then, many studies have led to the granting of the probiotic status to different LAB responsible for the fermentation of foods frequently associated with health benefits [5,6]. Many fermented foods are considered functional foods as they contain microorganisms, enhancing the overall health level of consumers [7]. Due to the positive effects exerted both by fermenting microorganisms and the products of their metabolism, LAB-fermented foods could represent a safe, unexpensive, and reliable tool in improving human health. This review highlights the health effects reported by the literature on LAB responsible for the fermentation of different foods, which could contribute to maintaining and promoting consumers’ health.

### 1.1. Lactic Acid Bacteria as Probiotics

Despite the evolution of the probiotic concept, the term probiotic has been linked to bacteria beneficial for the host health since Elie Metchnikoff’s observation that the regular consumption of dairy products fermented by LAB was associated with enhanced health and longevity in the elderly Bulgarian people [8]. Over the years, a considerable number of microorganisms was proposed as probiotics, with health-improving abilities. Most known probiotics belong mainly to the LAB group and Bifidobacteriaceae, while for others such as the yeast *Saccharomyces boulardii,* or *Enteroccoccus spp*., the use as probiotic has been debated for long due to their intrinsic opportunistic nature and the possibility to transfer antimicrobial resistance genes [9,10]. In contrast to the traditional probiotics, non-conventional, native gut microbiota bacteria have rapidly attracted much more attention for promoting health and therapeutic purposes, leading to the concept of Next-Generation-Probiotics (NGP) [11,12]. Because of the development in microbial culturing techniques [13], in the metagenomics and genomics technologies involved in sequencing [14] and editing of bacterial genome [15,16], the range of microorganisms considered for their potential positive effects on hosts health has nowadays broadened up, involving non-LAB genera like *Actinobacteria* (*Akkermansia muciniphila* is among many others an emerging star in the field), *Bacteroidetes*, *Firmicutes*, and *Verrucomicrobia* [12,17]. Despite this, LAB are still the most used health-related bacteria in food production. Due to their long history of safe use, LAB have been listed either as Generally Regarded as Safe (GRAS) at the strain level by the United States Food and Drug Administration (FDA), or as Qualified Presumption of Safety (QPS) at the species level by the European Food Safety Authority (EFSA). Thus, they can be used as food or food supplements [11], and confer to the fermented food functional characteristics, entailing a vast arsenal of aces in the hole in granting benefits to human health [7].

### 1.2. Health Effects of Probiotics

Health-related features ascribed to probiotic microorganisms are multifaceted. Probiotics are known for: (i) the production of valuable compounds, (ii) antagonist activity towards pathogenic bacteria, (iii) stimulation and regulation of immune response, and many other effects [2]. As they generally exert their effect starting from the intestine, probiotics should show: (i) good resistance to acids, and (ii) disaggregating effect of biliary salts, (iii) ability to colonize intestinal walls, (iv) compete for nutrients, and (v) remain alive in the harsh and selective conditions of Gastro-Intestinal (GI) tract [2]. By colonizing intestinal mucosa and interacting with the mucus layer, probiotics modulate immune response, improving defense to external attacks. Maintaining a constant presence in the gut, the immune system is stimulated, also leading to reduced severity of autoimmune aggressions and lowering allergic response, according to Rook and the “old friend theory” [4,5]. Indeed, in the colonic region, from the fermentation of digested material, they can produce antioxidants and anti-carcinogenic compounds, together with a series of molecules activating a signalling process between bacteria and intestinal epithelium [18]. This starts a cascade of effects that eliminate pathogenic and harmful microorganisms, thus creating a better environment and maintaining homeostasis [2]. To reach these goals, probiotics might be in a viable state and with an adequate amount [19]. However, some experimental evidence suggests a role exerted by non-viable or dead microbial cells in improving the health status of hosts, opening the door to the concepts of post-biotics and para-probiotics [6]. Post-biotics term is referred both to non-viable microorganisms present in the preparation and to soluble compounds released by probiotics after cellular lysis, comprising (i) short-chain fatty acids (SCFA), (ii) lactate, (iii) cellular wall components, and (iv) peptides [20]. Conte et al. reported using post-biotics from different lactobacilli as treatment to reduce the entrance of gluten proteins in CaCo-2 cells of patients affected with celiac disease [20]. Para-probiotics comprise non-viable microorganisms and the entire microbial fraction released after cellular lysis [21]. Sugawara et al., in an intervention study, showed an improvement in intestinal environment and functions after 3 weeks of consumption of a para-probiotics beverage containing non-viable cells of *Lactobacillus gasseri* [21]. Both viable and non-viable (or part of) cells can interact particularly in the intestinal epithelium through the stimulation of intracellular signalling pathways [22]. Many of these features have been described in LAB, which can produce different compounds, like bioactive sequences of peptides, sugars polymers, and fatty acids involved in boosting human health [11]. LAB can also produce organic acids, bacteriocins, hydrogen peroxide (H_2_O_2_), and nitric oxide (NO), that are active against pathogens [10]. Furthermore, during fermentation in the intestinal lumen, LAB also produces SCFA. These acids can be produced also by other microorganisms, for example: acetate can be produced by *Akkermansia muciniphila, Bacteroidetes, Bifidobacterium* spp., and *Clostridium* spp.; propionate by *Veillonella parvula*, *Bacteroides eggerthii*, *Bacteroides fragilis*, *Ruminococcus bromii*, and *Eubacterium dolichum*; and butyrate by *Faecalibacterium prausnitzii*, *Clostridium leptum*, and *Eubacterium rectale* [23]. SCFA are involved in different processes, for example butyric acids furnish metabolic energy to colonocytes and is studied for its effect in avoiding the development of cancer cells [22]. Propionate enhances gluconeogenesis and helps maintain glucose homeostasis in the organisms by increasing the expression of leptin, an anorectic hormone, in adipocytes [24]. Acetate is involved in the lipogenesis and synthesis of cholesterol [25].

### 1.3. Health Effects of Foods Fermented by LAB

In the past years, the consumption of probiotics was strongly recommended, and the involvement of positive microorganisms in the formulation of foods with a health claim was widespread. Nowadays, due to a more profound knowledge of the probiotics’ health effects and the mechanism behind them, it is possible to broaden the range of microorganisms involved in the formulation of functional foods. In some cases, LAB that are part of the spontaneous microbial population of one food, drive the beneficial effects to the host without being recognized (yet) as probiotics [26,27,28,29]. Positive effects connected to fermented foods have been empirically known for centuries. In many cultures, fermented foods are heritage foods and an integral part of local traditions, probably because fermentation was the only way to preserve foods [29]. Nowadays, regular consumption of fermented foods, especially lactic-fermented ones, has been reported to improve the immune system, reducing the probability of developing morbidities [27] due to a constant communication between bacteria and host immune system. This communication changes the microbial composition of the intestine, maintaining under control pathogenic microflora and meanwhile supporting beneficial microbes populations [30].

Among fermented foods, dairy products have been mainly associated with beneficial effects. This is partly due to the significant number of proteins available in the substrate for cellular duplication. During fermentation, because of acidification and microbial enzymes activity, proteins are denatured and lose their original conformation, releasing sequences of small peptides studied for their potential health-related effects. One of the most studied and regarded groups of bioactive peptides is represented by Angiontensin-1-Converting Enzyme (ACE) inhibitors. These bioactive peptides have been studied for their anti-hypertensive effect, and several guidelines suggest consuming fermented dairy products as a non-pharmacological way of controlling hypertension. Scientific evidence reported two main peptides as carriers of hypotensive effect: VPP (valine, proline, proline) and IPP (isoleucine, proline, proline) [29,31,32]. ACE inhibition occurs when ACE I is sequestered by the C-terminal sequence of ACE-inhibitors. In this way, ACE cannot convert angiotensin I in angiotensin II, a potent vasoconstrictor. Synthesis of angiotensin II leads also to degradation of bradykinin, a vasodilator; soaring blood vessels’ constriction; and dramatically increasing blood pressure [31,32].

Furthermore, LABs can produce exopolysaccharides (EPS), long sugars polymers formed by repeated units of mono- or oligosaccharides, that are gaining a lot of attention from the scientific community, due to their technological role [33], but also for their promising health benefits [34]. EPS can be divided in two macro-categories depending on the sugars presents in the main chain: (i) Heteropolysaccharides (HePSs) are polymers of different monosaccharides; (ii) Homopolysaccharides (HoPSs) are polymers of one sugar, repeated many times. In the latter case, HoPSs can be divided into glucans or fructans depending on the sugar composing the polymer chain, glucose, and fructose, respectively. Production of HoPS takes place outside microbial cells, mediated by membrane enzymes that hydrolyse and reassemble the sugars in a new EPS chain. By contrast, HePSs synthesis is more complex, and the chain contains more than one sugar moiety, normally being glucose, galactose, and rhamnose. Still, in different LAB’s EPS it is possible to find different sugars or other functional groups like acetyl and phosphate groups [35]. Normally, HePSs are associated with the modulation of host function, e.g., antioxidant effect or immune modulation, while HoPSs are associated with prebiotic properties, indicating how the conformation of these branched sugars and the monomeric composition influence the impact on the host [35,36]. The prebiotic effect exerted by LAB’s EPS is the subject of particular interest, because of the production of SCFA, gasses, and organic acids involved in the inhibition of noxious bacteria and the improvement of host’s metabolism [35]. EPS produced by LABs proved to be more effective in increasing the amount of Bifidobacteriaceae in the intestinal lumen with respect to inulin, the most used bifidogenic oligosaccharides. At the same time, an antagonist effect towards *Bacteroides* and Clostridia was shown. Gut microbiota is strongly affected by the presence of EPS in the intestinal lumen, especially by HoPSs, that result to be the most suitable substrates for fermentation, while HePSs are normally not fermentable, but their ability to modulate the immune system make them of capital importance in maintaining a general health status [36].

In fact, EPS are supposed to have antioxidant and immunomodulatory effects, as well as the ability to reduce cholesterol in the bloodstream and its absorption; anticancer and anti-diabetic effects are just some of the positive features that may be exerted. Furthermore, they also have a role in fighting the presence of harmful bacteria in the intestine, since they can disrupt biofilms, removing the protection of pathogenic microorganisms and exposing them to stresses and attacks. Different studies were carried out to explore these proposed effects for EPS. Still, it has to be considered that many of these experiments were carried out in vitro or with animal models, missing the confirmation from clinical trials on humans [36]. Some studies on animals pointed out the anti-cholesterolemic effect of EPS. This effect is based on increasing the high-density lipoprotein (HDL) ratio: total cholesterol with reduction of lipidic deposits in the bloodstream, especially in the aorta. In other experiments, it was observed that bile acids were scavenged by EPS, reducing in this way the amount of cholesterol present in the blood. This can be due also to the utilization of blood cholesterol to synthesize new bile acids, which are subsequently employed in digestion processes. Results are of course promising, even if the mechanism through which EPS lowers cholesterol content in the blood is still not precisely known [36].

Health effects of food fermented by LAB (Figure 1) are known and have been studied for a long time. Despite this, we do not yet know all the mechanisms of action and the secondary effects of LAB and their derived compounds. For many years, literature have focused on health effects of bacteria isolated and recognized as probiotics, but more recent studies shed light on the beneficial effects of bacteria involved in food fermentation that are not considered probiotics due to the non-complete compliance to probiotics guidelines. As an example, LAB proved to be useful in homeostasis both directly in the gut and indirectly utilizing pathways’ modifications that lead to an improvement of host health status [37,38].

## 2. Health-Related Effects of Different LAB Fermented Foods

### 2.1. Fermented Dairy Products

Milk is probably one of the first fermented food staples by mankind. Historically, the first fermentations happened accidentally due to unpasteurized milk’s tendency to spontaneously ferment due to the high level of nutrients and microbes [40,41]. From a biochemical point of view, fermentation is a complex combination of events. After lactose metabolism, different compounds are generated, such as: acids, ethanol, and carbon dioxide. The production of acids leads to a decrease of the pH, limiting the growth of negative microflora. Aroma compounds are also produced, increasing palatability and acceptance of foods and nutritional compounds like vitamins, minerals, bioactive molecules, and EPS [42]. Nowadays, after millennia of traditions and evolution of dairy art, fermented milk products represent about 20% of the total revenue generated by the fermented-foods markets all over the world. Production of fermented milks arose after 1950 when the demand for yoghurts and other similar products increased sensibly, attracting the attention of companies and consequently moving the production from a small-scale, in artisanal farms, to a mass production led by big multinationals [42]. Milks from different animals have become raw material for dairy fermentations. In fact, it is possible to find yoghurts, cheeses, and sour milks produced with cow milk, goat, sheep and horse milk as just examples in global markets. Even though dairy fermentations originally started from wild LAB present in milk, nowadays companies cannot rely anymore on spontaneous microflora, because of technological properties and possible health issues related to raw materials. For this reason, almost all industrially-fermented dairy products are produced with selected starters, or with back-slopping technique [42,43,44]. Fermented dairy products can be divided in different categories; in this review, for the sake of brevity, we focus only on fermented milks and cheese. Fermented milks are many and can be classified basing on: production techniques, the origin of milk, and other factors [45]. Since the variety of these products is humongous, considering traditional and industrial processes, novel fermented milks, and ones deeply rooted in archaic societies, we only consider the two most consumed and spread fermented milk products: yoghurt and kefir.

#### 2.1.1. Yoghurt

Due to its taste and versatility, yoghurt is one of the most consumed milk-derived products worldwide [46]. Like other dairy products, yoghurt is strongly recommended in diets, for its provided nutrients, like essential amino acids, and bioactive compounds, such as lactic acid, EPS, and liposoluble vitamins [40], which are otherwise rare and difficult to be introduced with the diet [47]. In a standard yoghurt’s serving, it is possible to find many useful nutritional compounds like (i) vitamins and minerals in a rapidly absorbable form [48]; (ii) bioactive peptides with many health-modulating effects [29,49]; (iii) branched-chain amino acids (BCAA) positively correlated with muscle growth and body maintenance [50]; (iv) mono- and poly-unsaturated fatty acids vehiculating liposoluble vitamins (A, E, K, and D); and (v) conjugated linoleic acid (CLA), known for the anti-carcinogenic activity and apoptotic induction in cancerous cells, as reported by different papers, especially towards breast cancer in vivo and in vitro [46,51,52,53,54]. All the listed compounds, or precursors, are already present in milk, but the fermentation process is essential to liberate this vast amount of positive health-related compounds in the matrix. Fermentation of milk to produce yoghurt is carried out by two specific LAB: *Lactobacillus delbrueckii* subsp. *bulgaricus* and *Streptococcus thermophilus,* even if other species can be added as a plus. Bacteria from yoghurt are known for making part of the so-called transient microbiota, since they usually cannot colonize the intestine. Despite this, the health contribution of yoghurt microflora should not be underrated. Kousgard et al. reported a clinical trial on patients affected with pouchitis and treated with a fecal microbiota transplant. In that study, four out of four patients with pouchitis symptoms remission regularly consumed yoghurt, while only one out of five patients with relapse issues consumed yoghurt on a daily basis [55]. The remission effect could also be correlated to the presence of organic acids produced by microorganisms, which contribute to fighting pathogenic microorganisms and maintaining a safer gut environment. Several dietary guidelines suggest the implementation of yoghurt in a healthy diet daily, also for lactose-sensitive people, due to the ability of the contained LAB species to improve this sugar digestion [56]. Different scientific papers focused on the utilization of probiotic fortified yoghurts in the management of type 2 diabetes. At the same time, Barengolts et al. in a meta-analysis of randomized controlled trials demonstrated that consumption of yoghurt can improve management of diabetes complications, reporting no difference between effects exerted by conventional and probiotic fortified yoghurts [57]. Kong et al. reported the utilization of yoghurt in combination with fruits and caloric restriction to fight non-alcoholic-fatty-liver-disease (NAFLD). Their data showed the ability of the diet intervention to modify the gut microbiota. An intimate relationship between gut and liver is already well known to the scientific community. In fact, results from that research paper highlighted how modifications in gut microbial population can retard or even prevent the start of different chronic diseases, like NAFLD, among others [58]. Liu et al. in clinical tests on mice reported traditional yoghurt being able to modulate intestinal microflora, repairing and avoiding dysbiosis that can negatively affect brain functions and behaviour. In fact, in transgenic mice modified to develop Alzheimer disease’ (AD) symptoms in early stage of life, yoghurt’s supplementation reduces the deposition of myeloid-beta plaques in brain cortex and hippocampus, event though it is highly correlated with the onset and development of AD disease. It derives that gut microbiota modulation, operated by ingestion of yoghurt, and its microbiome can help in reducing the issues connected with AD and cognitive function [59,60]. Considering all these health-related effects, yoghurt reveals to be a cost-effective way to introduce in the diet countless health-boosting compounds, it helps in the management of non-communicable disease, and is negatively associated with all-cause mortality [60,61,62].

#### 2.1.2. Kefir

It is one of the first fermented milks. Traditional kefir owes its longevity in human diet and traditions to its peculiar organoleptic characteristics and to an unconscious association with health benefits and life prolongation [63]. Traditionally, kefir is made by the action of kefir grains, in which are comprised LAB, Acetic Acid Bacteria (AAB), and yeasts enveloped in a slimy matrix composed of EPS and proteins [64]. Kefir can be defined as a “natural complex probiotic” because of the interaction between many different microorganisms, and it is supposed to exert anticarcinogenic, immunomodulatory, antiallergenic, antidiabetic, antistress, and antiasthmatic effects [65,66,67]. Kefir microflora depends not only on the inoculum of the grains, but also on external factors (light, temperature, kefir grains/milk ratio, agitation…), which can influence organoleptic features as well as bioactivities, favouring the growth of specific strains, while a core population always exists [68]. Health-related effects of kefir can be ascribed to the presence of bacteria, but also to bioactivities [69]. It can, for example, modulate gut microbiota and increase *Lactobacillus* and *Bifidobacterium*, while decreasing *Bacteroidetes* level in the intestine of patients affected with metabolic syndrome, leading to improvements in fasting glycaemia, reduction of inflammation signals, and blood pressure [70]. Modifications of the gut microbiota exerted by kefir’s bacteria are reported also by Yilmaz et al., who noticed in a randomized control trial that *Lentilactobacillus kefiri* LK9 was able to colonize the intestine of volunteers after 1 month of administration, resulting as present in faeces at 10^5^–10^6^ Log CFU/g. *L. kefiri* is also reported to inhibit other microorganisms associated with the start of pro-inflammatory chain events and gastrointestinal illness [71]. Kim et al. investigated the effect of kefir in reducing the incidence of obesity, induced by a high-fat diet (HFD) and NAFLD. In their experiments, results show a decrease of 60% of incidence of obesity in mice concerning control group, showing that a 0.2 mL supplementation of kefir reduces the effects of HFD and related NAFLD. Also, blood cholesterol and systemic inflammation, both induced by a fat-rich diet, were reduced by kefir supplementation. The mechanism of action in the reduction of obesity and related problems seems to be exerted by the cooperation of three different factors: LAB, yeasts, and EPS. In fact, kefir-derived bacteria can influence the gut microbiota directly by colonizing gut epithelium and indirectly by modifying pH of the intestinal lumen and inducing expression of genes that codify for useful enzymes. Reduction of pH creates a harsh environment for pathogenic and undesired microorganisms, but not for LAB that are normally used in acidic environments. In the same experiments, Kim et al. concluded that the introduction of probiotics derived from natural kefir is able to up-regulate peroxisome proliferator-activate receptor. This system plays a central role in beta-oxidation and reveals to be a fundamental drug helping in fighting NALFD [72,73]. Many studies in recent years focused on anti-cancer abilities of fermented foods, and kefir is one of the most investigated since its health-boosting effects have been known from the dawn of time. Anti-cancer activities exerted by kefir are mediated by different compounds, like bioactive peptides, EPS, and sphingolipids. The mechanism of action of these compounds seems to be bound to modulation of signalling pathways and of cells’ processes, e.g., cellular proliferation and apoptosis [65,74]. In a systematic review of the literature, Rafie et al. reported that according to the state of the art, the mechanism of the action exerted by kefir in inducing apoptosis is not fully understood yet, but it can be due to the formation of reactive oxygen species (ROS), mediated by peptides. The liberation of ROS in the cell creates damage and activates endonucleases that cleave DNA, creating an escalating apoptotic effect. ROS disrupt mitochondria, creating a cascade of events that leads the cells to death. This cascade effect seems extremely powerful since peptides from kefir are naturally positively charged, thus being electrochemically attracted by negatively charged components of cancerous cells. EPS contributes to apoptosis of cancerous cells, activating macrophages and T-lymphocyte. Moreover, regulation of genes expression seems to be involved in anti-tumour potential of kefir, as its consumption seems to up-regulate pro-apoptotic systems and down-regulate proliferations systems [75]. In their review, Rafie et al. reported the amount of kefir supplemented for the experiments, ranging from 200 µL to 5 mL, but, as all the listed experiments are in vitro on cancerous cells, the precise amount that has to be consumed to reach a positive effect needs to be further investigated [75]. Kefir was administered by Özcan et al. to postmenopausal women to improve quality of sleep and thus reduce mental disorders, like depression and stress accumulation. It is well known that the gut–brain axis is a highway, and what affect the guts, reflects on the brain [76,77]. In this sense, the beneficial effect of kefir reducing harmful microflora, improving motility, and modulating immune function helps to reduce sleeps disorders, depression, stress, and anxiety, thus increasing the quality of life. In this study, patients were supplemented with 500 mL of kefir daily, to drink half in the morning and the rest in the evening. The ingested amount is considerably high, but it has to be taken into account that the experiments were conducted in Turkey, where kefir consumption is traditionally rooted in the population [78]. Kefir was administered also to ovariectomized mice to study the effect of kefir’s peptide fraction on estrogenic deficiency-induced osteoporosis and evaluate in model systems prevention of menopausal osteoporosis. As already stated, in fact, kefir can modulate gut microbiota through different patterns, influencing many aspects of physiological processes like absorption of nutrients, hormone regulation, and metabolic processes. Moreover, through EPS of kefiran, kefir exerts a bifidogenic effect, increasing sensibly the amount *Bifidobacterium* in the guts, reducing the amount of pathogenic microflora (fungi, protozoa, viruses, and bacteria), due to the production of organic acids and bioactive peptides [79]. Modulation on the hosts exerted by kefir is also broadened by the promotion of fatty acids oxidation by increasing *Lactobacillaceae* population as well as *Kluyveromyces* spp. presence in the gut [72]. Kefir containing a natural probiotic, able to release SCFA in the media and the guts, contributes to bone formation and improves bone density [80]. Different studies focused on this topic both in animal and humans, confirming the effect of kefir in reducing bone loss, increasing bone density and elastic moduli of bones, and preventing fractures that may result in fatal ending for elderly persons. This effect is enhanced when combined with calcium-carbonate supplementation [81,82,83]. In the end, being so widespread, easy to use, and obtain, kefir looks like a treasure chest of positive effects for consumers.

#### 2.1.3. Cheese

Cheese is an umbrella term under which many products differentiated by production techniques, composition, environment, and microbial evolution find space. The combination of productive processes and microbiota are fundamental to differentiate products. For example, during fast ripening, the amount of lactose is reduced by microorganisms, leaving a reduced amount of lactose final product, making these cheeses a choice for lactose-sensitive individuals. On the other hand, during prolonged ripening, which can last for months and even years, lactose is completely consumed by LAB, making these cheeses an attractive source of dairy micro- and macro-nutrients for lactose-intolerant people. During the first stages of fermentation, bacteria consume carbohydrates, leaving just a fraction of indigestible oligosaccharides in the matrix that is proven to reach the intestine and exert prebiotics effect, stimulating positive microflora [84,85]. During early stages of ripening, lactose is rapidly degraded in lactate, by means of starter LAB. Lactate can then be metabolized by *Propionibacterium*, Clostridia, and Pediococci in propionate, butyrate, and formic acid, respectively [86]. Milk contains also citrate that is normally involved in LAB metabolism by citrate positive bacteria, mainly Lactococci [86]. Strains usually involved are *Lactobacillus lactis* ssp. *lactis* biovar *diacetylactis*, and *Leuconostoc mesenteroides*, which produce acetate, diacetyl, 2-butanone, and 2,3-butanediol [86]. Another important metabolism of LAB during ripening is proteolysis, resulting in the release in the matrix of branched-chain amino acids such as leucine, iso leucine, and valine; aromatic amino acids such as tryptophan, phenylalanine, and tyrosine; and sulfur-containing amino acid such as methionine. Peptides and ammino acids in cheese are often in an interesting bioavailable form [86]. During ripening time, small peptides are released by the action of enzymes, residual rennet activity, and LAB. A part of these peptides can be metabolized by LAB [87], and is well known for bioactivities, such as opioids, ACE-inhibitors, and immuno-stimulating activities. Some other peptides vehiculate minerals to the intestine and peripheric organs via blood transport [88]. There is an expanding body of evidence concerning a negative correlation between intake of dairy products and development of hypertension [89]. This anti-hypertensive effect seems to be correlated to the presence of calcium and small peptides with ACE-inhibitors activity, like IPP or VPP peptides [90,91]. Ripening of cheese is positively correlated with these bioactive peptides, which are normally present in cryptic form inside caseins. In a double-blind study, Crippa et al. fed Grana Padano, a long ripened Italian cheese, to 30 patients with hypertension issues and reported a significant decrease in systolic and diastolic blood pressure after 2 months of administration of 35 grams of grated cheese per day. The decrease of blood pressure was in the order of −4.8/3.5 mmHg, which is interesting considering that a reduction of 3 mmHg can reduce the risk of heart attack and failure of about 13% [92]. In recent years, cheese-isolated probiotics have gained attention due to their ability to produce a variety of bioactive compounds like SCFA from the fermentation of non-digestible carbohydrates [93]; their antimicrobial effect towards pathogenic microflora; as well as their ability to improve immune response, reduce serum cholesterol level, and alleviate diarrheic symptoms [94]. Recently, literature focused on compounds with the ability to modulate mood [95,96]. One of the most studied mood-modulators is γ-aminobutyric acid (GABA). GABA is a non-protein amino acid derived from decarboxylation of glutamate [87] and is one of the main inhibitory neurotransmitters in the central nervous system of mammalians. Studies showed its involvement in managing stress, influencing behaviour and personality, and hypotensive and anti-diabetic properties [95,97]. Moreover, its effect was also noticed in preventing depression and helping in the treatment of alcoholism by activating specific receptors and increasing lymphocyte counts [87]. Strains able to produce GABA during fermentation of milk are *Lacticaseibacillus paracasei*, *Lentilactobacillus buchneri*, *L. delbrueckii* subsp. *bulgaricus*, *Lactiplantibacillus plantarum*, *Levilactobacillus brevis*, *Lacticaseibacillus rhamnosus*, and *Lactococcus lactis* [95,98]. Cheese seems to exert a protective effect towards these bacteria, due to the high fat content that protects bacteria and allows them to reach the intestine, where they can exert multiple positive effects [98]. Knowing this, the introduction of cheese, especially long ripened ones in the diet, allows the introduction of numerous positive compounds like bioactive peptides, minerals, liposoluble vitamins, organic acids, and other antimicrobial compounds, together with a positive and stress-resistant microflora (Table 2). Moreover, cheese can convey mood modulators to the hosts, helping in the management of stress and altered mood states.

### 2.2. Vegetable Fermented Products

Since ancient times, the fermentation of vegetables has also been practiced by mankind, as proved by a long history of traditional products spread all over the world. Vegetables are mainly fermented by LAB both spontaneously and by means of inoculum and back-slopping [103,104]. Among these lacto-fermented vegetables are fermented cabbage (kimchi and sauerkrauts), fermented leaf (gundruk) and pickles (cucumber, chillies, capers and others). Many of the positive features related to fermented vegetables are derived from the effects of acids and fermentation, which, as a consequence of fermentation, change their form to become more bioavailable, thus increasing their effect and elimination of anti-nutritional compounds [105]. In this review, we focus on the two main products derived from cabbage fermentation, representing a widely consumed staple in western and eastern areas of the world: sauerkrauts and kimchi. Fermentation of vegetables has as primary effect of increasing the shelf-life of food. Moreover, it allows to ameliorate the intake of nutrients like fiber, vitamins, and minerals. This effect is particularly useful since it permits the introduction of these micronutrients in periods when vegetables are unavailable. In a recent review, Bousquet et al. tried to find a relation between decease due to COVID-19 and diet of populations, focusing on the consumption of sauerkrauts. From their data analysis emerged how in the areas where the consumption of sauerkrauts is higher the number of deaths is slightly lower. Data anyway do not seem to be correlated and many other factors and bias contribute to the obtained results, thus further studies are needed to confirm any link [106].

#### 2.2.1. Sauerkraut

Sauerkrauts are the product of cabbage fermentation (*Brassica oleracea* var. *capitata*). Sauerkraut manufacturing can be carried out following spontaneous fermentation or fermentations guided by selected and specific bacteria [107]. During fermentation, the composition of the product changes and, at the end, aside from macronutrients, it is possible to find a good amount of fiber, vitamin C, organic acids (lactic, acetic, malic and succinic), SCFA (propionic acid), ethanol, and acetaldehyde. Due to the knowledge about bioactive compounds present in fermented foods, in recent years, many efforts were made to improve the general quality of fermented vegetables while creating a product rich in bioactive compounds. For example, the utilization of a nisin-resistant strain of *Leuconostoc mesenteroides* in combination with a nisin-producer strain of *L. lactis* allowed obtaining a product with a suppressed native microflora [107]. Also, *Leu. mesenteroides* in combination with *Pediococus dextrinicus* showed a good potentiality to produce bioactive enriched foods. *Latilactobacillus sakei* showed a predominance in this feature since its utilization in vegetable fermentations allows the obtaining of foods with three times the concentration of bioactive compounds concerning any other studied bacterial strains [107]. Standardization of the product is of course a feature researched by companies. Despite this effort to standardize the products, aiming to use only selected microorganisms, it has to be considered that a reduced microflora diversity could lead to products with decreased bioactivities and a lower release of post- and para-probiotics in the final product [108,109,110]. Thus, aiming to obtain a safe and health-contributing product, it is important not to underestimate the potential contribution of the autochthonous microflora in the fermentation process [111]. Since many reports suggest that regular consumption of this product can lead to the intake of a considerable amount of healthy bacteria (>10^6^ log CFU/g), recent studies have focused their attention on the isolation of LAB from sauerkrauts. Strains of *Lactiplantibacillus paraplantarum*, *L. brevis,* and others *Lactobacillus* strains isolated from sauerkrauts showed adhesion to Caco-2-cells and inhibitory activity towards pathogenic microorganisms [107]. Nielsen et al. [112] reported that the effect of sauerkraut consumption on irritable bowel syndrome (IBS) affected a patient and reported that consumption of sauerkrauts, both fresh and pasteurized, led to a reduction of symptoms after 6 weeks, with a change in microbial composition of faecal matters of participants. Also, the high presence of dietary fibers seems to be involved in alleviating IBS symptoms [112]. Cabbages are also rich in phytochemicals with multiple possible bioactivities, but these compounds, mainly glucosinolates, are normally not bioavailable in the fresh product. Hydrolysis of glucosinolates leads to release of isothiocyanates, thiocyanates, epithionitriles, nitriles, and indolic compounds, all recognized for their valuable health boosting activities. Like many other fermented foods, sauerkrauts show antitumoral properties, exerted by activating enzymes that eliminate xenobiotics and increasing apoptosis of cancerous cells [107]. Specifically, indole-3-carbinol (I3C) is deeply investigated since it was shown to exert inflammation-modulating effects, promote cells proliferation, and inhibit tumour invasion in different tissues [107]. The presence of vitamins and organic acids gives to sauerkraut a powerful antioxidant feature, but it is also related to reduced inflammation, atherothrombosis, and increased human system efficiency in neutralizing reactive oxygen species. Antioxidant activities are also connected to reduced oxidative damage at the expense of DNA, which can also be due to indolic compounds’ ability to scavenge chemicals, avoiding damages to DNA and other structures [113,114]. Fermentation enriches sauerkraut with a group of enzymes called Mono Ammino Oxidase Inhibitors (MAOIs), inhibiting Mono Ammino Oxidase (MAOs), which are a family of enzymes involved in arising depressive states, anxiety, obsessive-compulsive disorder, and development of Parkinson’s disease [115]. The administration of sauerkraut was also studied in fighting IBS. In a pilot study, Nielsen et al. fed 34 volunteers with pasteurized and unpasteurized sauerkraut to evaluate the reduction in abdominal discomfort and problematics bound to IBS. From the results, it emerged that administration for 6 weeks of unpasteurized sauerkraut and 8 weeks of pasteurized sauerkraut can sensibly reduce abdominal discomfort and negative effects of IBS. Despite the difference in number of live bacteria, this similarity in results can be due to the natural composition of sauerkrauts, rich in glucosinolates and complex carbohydrates, acting as fiber in the intestine. In this optic, fermentation of sauerkrauts leading to glucosinolates breakdown can increase the bioactivity of this fermented food. Also, cells breakdown and liberation in para-probiotics media can contribute to the health-related positive effects of fermented cabbages [112]. Further experiments in this field should consider unpasteurized cabbage to estimate precisely the effect of fermentation with respect to unfermented product [116]. When talking about sauerkraut, many sources refer to its potentiality as a source of fiber and healthy compounds, forgetting about the presence of an abundant and vital LAB microbiota, mainly deriving from spontaneous fermentations that select microorganisms with an increasingly harsh environment. These bacteria have increased possibilities to reach the gut and colonize intestine walls, where they can exert positive effects modulating the microbiota and immune response. As mentioned above, during recent years, literature has explored the idea that microorganisms’ viability is not mandatory to exert probiotic effects. Cell wall material, cytosol compounds, and genetic information released after cells death are enough to vehiculate positive features. In this new post- and para-biotic field, sauerkraut has found a niche, since industrial productions require a pasteurization step, resulting in the death of live cells but not hampering the beneficial effects of wild LAB. Sauerkrauts are one of the most studied fermented vegetables, in fact suggestions about the introduction of sauerkraut in the diet are easy to find in the literature, even though intervention studies and dietary supplementation with this fermented food are still lacking and further investigation is surely needed. The literature reveals that sauerkraut possess a vast array of health effects related to glucosinolate compounds and microbial contribution in terms of microbiome, para-, and post-biotics. Despite esethese incredibly appealing features, there is a possible presence of biogenic ammines [117], while paying attention to microbial populations, since some harmful bacteria can survive to harsh conditions that arise during fermentation.

#### 2.2.2. Kimchi

Kimchi is the most produced and consumed lacto-fermented vegetable of Korea and its national product. It is often made by natural fermentation of Napa cabbage and other ingredients like onion, garlic, chillies, and fish sauce; their addition is fundamental in helping to control pathogenic and harmful microorganisms, allowing the growth of the beneficial ones. It is mainly *Leu. mesenteroides* that creates the acidic and anaerobic environment adaptable to the growth of more acid-resistant bacteria like *L. brevis* and *L. plantarum*. Kimchi is considered a natural functional food for the high presence of dietary fibers, minerals, vitamins, capsaicin, organic acids, polyphenols, and fermentation by-products (organic acids, bacteriocins, and others). The presence of these compounds is reported in scientific literature to produce positive effects on the health of consumers. Presence of a wild and acid-resistant microbiota is connected to the lowering of pH in the intestinal lumen and in the faeces, which is connected to a better microbiota, with an increased count in LAB and *Bifidobacteria* and a lower level of harmful and pathogenic microorganisms. Kimchi is studied, especially in the most recent literature, for its ability to modulate gut microflora, and Park et al. studied the effect of kimchi to exert an anti-obesogenic effect on the microbiome, starting from the assumption that many factors can cause obesity, such as an unbalanced diet; genetic factors; and unhealthy gut microflora resulting in modifying energy intake and accumulation in the adipocytes, increasing obesogenic effect [118]. The intestinal microbiota is more than the sum of its part, it is an organism able to live in symbiosis between the same parts composing it and ourselves. Due to microbial diversity, long-term stability, ease of use, and domestic preparation, kimchi was taken into account to modify gut microbiota [117], helping pathologically obese subjects, normalizing their lipid levels and modulating their microbiome [119,120,121]. Results of these experiments highlighted that supplementation with kimchi in mice fed with an HFD cannot significantly decrease weight gain, with respect to mice fed with just an HFD, indicating that the number of calories introduced is the main factor in weight gain [118]. The introduction of kimchi anyway showed a reduction in blood glucose, triglycerides, and high- and low-density lipoproteins with respect to mice fed with only HFD [122]. Even if the total weight gain was not significantly decreased by kimchi in HFD mice, other indexes like total body fat gain, liver weight, and adipocytes’ dimensions and counts were lowered by the administration of kimchi. Also, gut microflora resulted modulated by the administration of kimchi, *Akkermansiaceae*, *Coriobacteriaceae*, and *Erysipelotrichaceae*, which are normally related to HFD and consequently to obese subjects, were lowered in mice fed with kimchi, while the abundance of *Muribaculaceae,* negatively correlated with obesity, increased in kimchi-fed mice [118]. Kimchi, due to the high presence of fibers and nutritional compounds was also studied as a solution to cope with prediabetics patients [123]. Prediabetics are subjects who have blood glucose higher than unaffected subjects, but not high enough to be considered properly diabetics, and are strongly subject to develop this issue later, due to unhealthy lifestyle and diet [123]. Fortunately, a change in dietary and lifestyle habits can slow, and in some case even stop, the progression of prediabetes into diabetes. In an intervention study, An et al. administered 100 grams of kimchi per meal to 21 prediabetic volunteers for 2 weeks, followed by a 4-week washout period. From anthropometric parameters after regular consumption of kimchi, it emerged that insulin sensitivity and resistance and blood pressure were positively affected by introduction of this fermented product. Also, the participants’ body mass index (BMI) and weight decreased significatively, together with waist circumference, which is strongly bound to insulin resistance. The consumption of kimchi thus revealed to be a strong ally in fighting the onset of diabetes [123,124]. Being rich in anti-microbial compounds, produced mainly by an active and resistant positive microflora, kimchi is employed from centuries as “medicine” food and can be ascribed in the functional foods group. Functional foods are “foods or dietary components that may provide a health benefit beyond basic nutrition” [125]. Some studies focused on the utilization of kimchi to fight infections by *Helicobacter pylori*, which is a well-known contributor to the development of peptic and perforative ulcers and one of the recognized class I carcinogens [124,126]. The high level of antioxidants, vitamins, and the presence of other phytochemicals, together with the reduction of ingestion of other harmful products prove to be an effective, cheap, and easy way to control the development of *H. pylori* and help eradicate it. During *H. pylori* infection, the body reacts by increasing the expression of pro-inflammatory genes to fight the attack, but the maintenance of an inflammation state for too long in the body can lead to DNA damage through oxidative stress and disruption of cell life cycle. In this way, gastric and intestinal carcinogenesis is facilitated. Therefore, the introduction into the diet of a food possessing antioxidant and anti-inflammatory properties such as kimchi results as being useful. In the optic of cancer’s prevention, kimchi has been employed as a tool to fight the development of colon-rectal colitis-associated cancer in patients chronically affected with IBS. From the experiments led by Han et al., it emerged that a special formulation of kimchi supplemented to mice, containing pear extracts and see tangle juice, prevents cancer formation by means of inflammasome reduction [127], resulting in anti-inflammatory and anti-oxidant effects, cytoprotective ability, and reduced proliferation of harmful microorganisms due to induction of apoptosis. In the same experiments, it emerged that the introduction of unfermented kimchi does not exert the same protective effect, accelerating the formation of cancers in the gut instead. This highlighted once more that the development of the native microbiota of the vegetable represents the real game changer in health-related effects of fermented foods [127]. Ordinarily, to cope with inflammation and ulcers, anti-inflammatory drugs are prescribed, and this can lead to resistance phenomena and to reduction of positive microbiota sensitive to drugs. Ingestion of kimchi could be a strong ally, due to ease of use, stability, and of course due to the introduction of a series of nutritional compounds exerting a plethora of positive health-related effects. Kimchi can play an important role also in maintaining under control the degeneration of several chronic diseases like IBS, Crohn’s disease, and infections due to external attacks or unhealthy eating habits [128,129,130]. However, it has to be considered that, as for other spontaneous fermented foods, kimchi contains a high level of salt involved in the formulation to control negative microflora. In some studies, the kimchi’s supplementation was in fact limited to around 100 g of fermented food [123], matching nutritional suggestions for salt introduction to the diet, while other studies increased to 210 g per day the administration of kimchi, which provides a salt content higher than what the guidelines suggest [131]. All this considered, the introduction of kimchi as a regular meal or side-dish also in Western countries could help populations to control the development of gastro-intestinal issues (Table 3).

## 3. Conclusions

The aim of this review was to highlight the ability of LAB involved in food’s fermentation to exert beneficial effects on human health. Regular ingestion of foods fermented by LAB in the diet can be a great help, due to the introduction of bioactive compounds that are released during fermentation and become available during digestion. It is well established that the ingestion of LAB-fermented foods can modulate the gut microbiome in its functionality and response to stress and attacks, both due to the presence of health-related LAB species and their metabolites produced during fermentation. LAB’s ability to produce bioactive peptides, vitamins, organic acids, bacteriocins, signalling molecules (NO), and antimicrobial compounds (H_2_O_2_) plays a fundamental role in promoting and maintaining a health status in consumers of LAB-fermented products. Despite the need of a higher amount of in vivo studies on a wider population and considering also the possible interaction among different fermented foods contemporaneously introduced, the pieces of evidence reported in the literature so far suggest that higher ingestion of LAB-fermented foods in the diet, daily, could contribute to a healthy lifestyle and in the maintenance of organisms functions and health.

## Figures and Tables

**Figure 1 foods-10-02639-f001:**
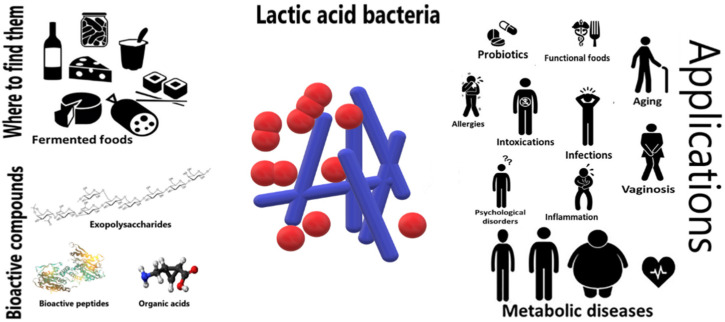
Scheme of LAB bioactive compounds and health-related effects, adapted with permission from [39] Copyright © 2018 George, Daniel, Thomas, Singer, Guilbaud, Tessier, Revol-Junelles, Borges and Foligné.

**Table 1 foods-10-02639-t001:** Food’s LAB groups and characteristics.

Families		Genera Frequently Found in Foods	CO_2_ from Glucose	Growth at 10	Growth at 45	Growth in 6.5% NaCl	Growth in 18% NaCl	Growth at pH 4.4	Growth at pH 9.6
*Carnobacteriaceae*	Rods	*Carnobacterium*	-	+	-	-	-	-	-

*Enterococcaceae*	Cocci	*Enterococcus*	-	+	+	+	-	+	+
		*Tetragenococcus*							

*Streptococcaceae*		*Streptococcus*	-	+	-	+	+	-	+
		*Lactococcus*	-	-	+/- *	-	-	-	-
*Lactobacillaceae*	Cocci	*Leuconostoc*	+	+	-	+/- *	-	+/- *	-
		*Oenococcus*	+	+	-	+/- *	-	+/- *	-
		*Pediococcus*	-	+	+/- *	+/- *	-	+	-
	Rods	*Lactobacillus*	+/- *	+/- *	+/- *	+/- *	-	+/- *	-
		*Lacticaseibacillus*	-	+	+/- *	+/- *	-	+	-
		*Lactiplantibacillus*	-	+	+/- *	+	-	+	-
		*Furfurilactibacillus*	+	+	-	+	+/- *	+	-
		*Fructilactibacillus*	+	-	+	+	-	+	-
		*Levilactobacillus*	+	+	-	+	+/- *	+	-
		*Limosilactobacillus*	+	-	+/- *	+/- *	+/- *	+	+/- *
		*Latilactobacillus*	-	+/- *	+/- *	+	+/- *	+	-
		*Lentilactobacillus*	+	+	+	+	-	+	+/- *
		*Weissella*	+	+	-	+/- *	-	+/- *	-

* Response may vary according to strains. “-“ absence of the tract. “+” presence of the tract.

**Table 2 foods-10-02639-t002:** Health-related effects of fermented dairy products.

Health Effects	Specific Effects	Fermented Food	Microorganisms	Reference
Reduce initiation and progression of cronic disease:		food ingredients, including living microbial cells	*Lactobacillus* and *Lactococcus* genera	[99]
	Musculoskeletal disorders			
	Cardiovascular diseases			
	Mental health pathologies			
	Type 2 diabetes			
Production of Bioactive peptides:		Milk-derived foods (Fermented milks, Cheese, yoghurt, kefir)	*Lactobacillus* and *Lactococcus* genera	[99]
	Satiety regulation			
	Antimicrobial			
	Anti-carcinogenic			
	Anti-thrombotic			
	Mineral absorption			
	Hypotensive			
	Anti-inflammatory			
	Stress relief			
	Aids relaxation and sleep			
	Reduces symptoms of psoriasis			
	ACE-inhibitors			
Amelioration of glucose metabolism		LAB-fermented foods, especially fermented milks	GRAS Lactic acid bacteria	[100]
Amelioration of glucose intollerance symptoms		LAB-fermented foods, especially fermented milks	GRAS Lactic acid bacteria	[100]
Reduce severity of infections		LAB-fermented foods, especially fermented milks	GRAS Lactic acid bacteria	[100]
Reduce burden of IBS		LAB-fermented foods, especially fermented milks	GRAS Lactic acid bacteria	[100]
Anti-anxiety effect		LAB-fermented foods, especially fermented milks	GRAS Lactic acid bacteria	[100]
Reduction of serum cholesterol level				[101]
Production of B’s group vitamines		Fermented milks, Yoghurts, Fermented Soymilk, Kefir	*L. casei, Bifidobacterium infantis, L. plantarum*...	[101]
Production of GABA				[101]
	Antidiabetic, blood pressure	Fermented milk, Fermented soy milk, Yoghurt	L. casei Shirota, S. salivarius, L. plantarum, L. brevis	
Production of conjugated linoleic acid				[101]
	Cholesterol lowering	Cheddar cheese, Buffalo cheese, Fermented buffalo milk, Yoghurt	L. lactis, L. rhamnosus, S. thermophilus, B. bifidum	[101]
Exopolysaccharides production				[101]
	Immunostimulatory	Yoghurt, Cheddar cheese, Turkish cheese, Kefir, Fermented ice-cream	*L. bulgaricus, L. mucosae, P. freudenreichii, L. lactis, B. longum*	
	Hypocholesterolemic			
	Microbiota modulation			
	Immune modulation			
Bacteriocines production		Camembert/Semihard cheese, Cheddar, Yoghurt, Munster cheese	*L. lactis, L. acidophilus, P. acidilactici*	[101]
Alleviate constipation		Yoghurt	*B. animalis* subsp *lactis* DN-173010, *L. casei* subsp Shirota	[102]
Reduce eczemas		fermented milk		[102]
Antibiotic-associated diarrhea		Fermented drink, yoghurt	*Lactobacillus casei* DN-114001	[102]
Prevention of pediatric diarrhea		Fermented drink, yoghurt	*Lactobacillus casei* DN-114002	[102]
Prevention and help healing from respiratory infections		Fermented drink, yoghurt	*Lactobacillus casei* DN-114003	[102]
Fights infections				[102]
	H. pylori infection	Fermented oat gruel in fruit drink	*L. plantarum* 299v (DSM9843)	
	Clostridium difficile infection	Fermented drink	*L. acidophilus* CL1285 + *L. casei* Lbc80r + *L. rhamnosus* CLR2	
Improves microbiota		Yoghurt	*L. acidophilus* + *B. animalis* subsp *lactis*	[102]

**Table 3 foods-10-02639-t003:** Health-related effects of fermented vegetables products.

Health Effect	Specific Effect	Fermented Food	Microorganisms	References
Antioxidant				
	Carotenoids modified by fermentation	Kimchi and Sauerkraut, Soybean,	*W. koreensis, L. brevis, Leu. gelidum*	[132]
Reduction of cronic diseases		Tomato Juice, Leek, Carrots,	*Leu. mesenteroides, L. plantarum, W. Confusa,*	
	Cardiovascular disease	Fennels, Onions, Pomegranate	*L. delbrueckii subsp. lactis, B. thermophilum*	[132]
	Cancer	Pear juice, Pineapple juice, Apple,		
	Diabetes	Quince, Grape, Kiwifruit		
	Alzheimer			
	Cataracts			
	Age-related functional declines			
Hypoglycemic				[132]
Anti-inflammatory				[132]
Hypolipidemic				[132]
Immunomodulatory				
Anti-microbic			*Lactobacillus* and *Lactococcus* genera	[132,133]
	Eliminate *H. pylori*			[133]
Reduction of anti-nutritional compounds		Fermented legumes and cereals	*Lb. plantarum* and other LAB	
Increase of nutritional density		Every fermented vegetable	Generic LAB	
	Breakdown of complex carbohydrates			
	proteolysis			
Glucosinolates breakdown				[132]
	Increase antioxidant activity	Brassica vegetables	Generic LAB	
	Ameliorate metabolic syndrome	Brassica vegetables	Generic LAB	
Production of SCFA		Every fermented vegetable	Generic LAB	[132]
Anti obesogenic effect		Every fermented vegetable	Generic LAB	
	Lower obesity incidence			
	Direct anti-obesogenic effect			
Prebiotic effect	Production of EPS		*W.confusa and W. hellenica, Lactobacillus, Lactococcus, Leuconostoc, Pediococcus* and *Weissella*,	[132]

## Data Availability

Not applicable.

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
