# Peer review of "Eating Fermented: Health Benefits of LAB-Fermented Foods"

_foods, 2021, doi:10.3390/foods10112639_

Round 1

Reviewer 1 Report

Better to have a general "Introduction" section with the aims/objectives of this review. 

It would be good to have a Table explaining/summarizing potential and proven health benefits of fermented dairy and vegetable products with relevant references 

Author Response

Reviewer 1:

  • Better to have a general "Introduction" section with the aims/objectives of this review. 

We thank the reviewer for the comment. We have created an “introduction” section with a general escursus on the topic, reporting our work's aim and objectives more clearly at line 30-61.

  • It would be good to have a Table explaining/summarizing the potential and proven health benefits of fermented dairy and vegetable products with relevant references 

We thank the reviewer for the kind suggestion. As suggested, 2 tables summarizing the health effects of dairy and vegetable fermented products have been added to the text.

Reviewer 2:

  • Fascinating manuscript however, the authors need to carry out major restructuring to be readable. First, it lacks an introduction. Second, we require at least 2 tables comparing different parameters and the health effects of LABs. References are written in the wrong form. Ref. 96 ios wrong concerning first names. Moreover, there are some recent papers on LAB which should be included. Authors need to carry out a recent literature review.

We thank the reviewer for the kind comments. According to what is suggested we included 2 tables in the text summarizing health-related effects of dairy and vegetable products. We also checked the references list x corrected reference’s form and checking the literature for the most recent articles.

  • The topic is relevant and interesting but not very innovative. However, it adds to the subject area compared with other published material since it is considered as advances. Paper is well written and has a lot of references and conclusions are consistent with the evidence and arguments presented addressing the main question posed.

We thank the reviewer for the positive comment

We addressed the text according to what was suggested by the reviewer all along with the text.

Lines 369-377: Explain how lactate and acetate that could produce from lactose. This is mainly because of the L. lactis and non specific lactic acid bacteria in milk. Then explain how it could produce acetate & diacetyl, 2-butanone and 2,3-butabediol due to S. diacetalyticus, L. lactis and L. cremoris like organisms.

We addressed the text according to the comment of reviewer 1, indicating how lactate, acetate and other products can be produced from lactose and citrate

Lines 378-382: Is this true? During ripening of cheese due to breakdown of proteins it could produce branched chain amino acids such as leucine, iso leucine and valine; aromatic amino acids such as tryptophan, phenylalanine and tyrosine and sulfur containing amino acid such as methionine. Authors need to check these information and include what could exactly produce during cheese ripening in general

 We thank the reviewer for the comment, and we have addressed the text according to reviewer 2 comment, expanding what reported concerning the production of branched and sulphuretted amino-acids

Line 443: is or Sauerkrauts

 We thank the reviewer for the comments, and we correct the text accordingly, now the sentence is: Sauerkrauts are the product of cabbage fermentation

Line 528: beneficial

We thank the reviewer for the comment, the sentence was corrected accordingly, now the sentence is: “allowing the growth of the beneficial ones”

Line 558: Not scientific, due to presence of high amount

We thank the reviewer for the comment, the sentence was corrected accordingly, now the sentence is: “Kimchi, due to the high presence of fibers and nutritional compounds”

Line 572-574: Introduce functional food concept here.

We thank the reviewer for the comment, the concept was addressed accordingly, now the reported sentence is: kimchi is employed from centuries as “medicine” food, and can be ascribed in the functional foods group. Functional foods are “foods or dietary components that may provide a health benefit beyond basic nutrition”

Lines 585-587: Rephrase this sentence

We thank the reviewer for the comment. The sentence was rephrased accordingly, and now is reported: “In the optic of cancer’s prevention, kimchi was employed as a tool to fight the developing of colon-rectal colitis-associated cancer in patients chronically affected wit IBS”

Conclusions: Remove references from the conclusions and state only the conclusions based on the above review information

We thank the reviewer for the comment. The “Conclusions’” paragraph was rephrased accordingly

Reviewer 2 Report

Interesting manuscript however the authors need to carry out major restructuring in order to be readable. First, it lacks an introduction. Second, we require at least 2 tables comparing different parameters and the health effects of LABs. References are written in the wrong form. Ref. 96 ios wrong with regard to first names. Moreover, there are some recent papers on LAB which should be included. Authors need to carry out a recent literature review.

The topic is relevant and interesting but not very innovative. However, it adds to the subject area compared with other published material since it is considered as advances. Paper is well written and has a lot of references and conclusions are consistent with the evidence and arguments presented addressing the main question posed.

Author Response

(The authors gave the same response as above.)

Round 2

Reviewer 2 Report

Authors have revised sufficiently according to our instructions

Author Response

Authors have revised sufficiently according to our instructions

We thank the reviewer for the comment. We also have checked the english all along the text and correct it according with the suggestion of required English changes